# Spelling interface using intracortical signals in a completely locked-in patient enabled via auditory neurofeedback training

Ujwal Chaudhary [1,8✉], Ioannis Vlachos[2,8], Jonas B. Zimmermann [2,8✉], Arnau Espinosa [2], Alessandro Tonin[2,3], Andres Jaramillo-Gonzalez [3], Majid Khalili-Ardali [3], Helge Topka[4], Jens Lehmberg[5], Gerhard M. Friehs[6], Alain Woodtli[2], John P. Donoghue[7] & Niels Birbaumer[3✉]

Patients with amyotrophic lateral sclerosis (ALS) can lose all muscle-based routes of communication as motor neuron degeneration progresses, and ultimately, they may be left without any means of communication. While others have evaluated communication in people with remaining muscle control, to the best of our knowledge, it is not known whether neural-based communication remains possible in a completely locked-in state. Here, we implanted two 64 microelectrode arrays in the supplementary and primary motor cortex of a patient in a completely locked-in state with ALS. The patient modulated neural firing rates based on auditory feedback and he used this strategy to select letters one at a time to form words and phrases to communicate his needs and experiences. This case study provides evidence that brain-based volitional communication is possible even in a completely locked-in state.

[1] ALS Voice gGmbH, Mössingen, Germany. [2] Wyss Center for Bio and Neuroengineering, Geneva, Switzerland. [3] Institute of Medical Psychology and Behavioral Neurobiology, University of Tübingen, Tübingen, Germany. [4] Department of Neurology, Clinical Neurophysiology, Cognitive Neurology and Stroke Unit, München Klinik Bogenhausen, Munich, Germany. [5] Department of Neurosurgery, München Klinik Bogenhausen, Munich, Germany. [6] Neurosurgery Department, European University, Nicosia, Cyprus. [7] Carney Brain Institute, Brown University, Providence, RI, USA. [8] These authors contributed equally: Ujwal Chaudhary, Ioannis Vlachos, Jonas B. Zimmermann. ✉email: chaudharyujwal@gmail.com; jonas.zimmermann@wysscenter.ch; niels.birbaumer@uni-tuebingen.de

Amyotrophic lateral sclerosis (ALS) is a devastating neurodegenerative disorder that leads to the progressive loss of voluntary muscular function of the body[1]. As the disorder typically progresses, the affected individual loses the ability to breathe due to diaphragm paralysis. Upon accepting artificial ventilation and with oro-facial muscle paralysis, the individual in most cases can no longer speak and becomes dependent on assistive and augmentative communication (AAC) devices[2,3], and may progress into the locked-in state (LIS) with intact eye-movement or gaze control[4,5]. Several invasive[6–10] and non-invasive[11–16] brain-computer interfaces (BCIs) have provided communication to individuals in LIS[17–20] using control of remaining eye-movement or (facial) muscles or neural signals. Once the affected individual loses this control to communicate reliably or cannot open their eyes voluntarily anymore, no existing assistive technology has provided voluntary communication in this completely locked-in state (CLIS)[17–20]. Non-invasive[11–16] and invasive[6–10] BCIs developed for communication have demonstrated successful cursor control and sentence formation by individuals up to the stage of LIS. However, none of these studies has demonstrated communication at the level of voluntary sentence formation in CLIS individuals, who lack stable and reliable eye-movement/muscle control or have closed eyes, leaving the possibility open that once all movement - and hence all possibility for communication - is lost, neural mechanisms to produce communication will concurrently fail. Several hypotheses have been formulated, based on the past BCI failures, to explain the inability of ALS-patients in CLIS to select letters to form words and sentences ranging from extinction of intentions[21] related to protracted loss of sensory input and motor output, cognitive dysfunction, particularly when it occurs in association with fronto-temporal degeneration. A successful demonstration of any BCI enabling an individual without reliable eye-movement control and with eyes closed (CLIS) to form a complete sentence would upend these hypotheses, opening the door to communication and the investigation of psychological processes in the completely paralyzed ALS patients and probably also other disease or injury states leading to CLIS.

Here, we established that an individual was in the CLIS state and demonstrated that sentence-level communication is possible using a BCI without relying upon the patient's vision. This individual lacked reliable voluntary eye-movement control and, consequently, was unable to use an eye-tracker for communication. The patient was also ultimately unable to use a non-invasive eye-movement-based computerized communication system[22]. To restore communication in CLIS, this participant was implanted with intracortical microelectrode arrays in two motor cortex areas. The legally responsible family members provided informed written consent to the implantation, according to procedures established by regulatory authorities. The patient, who is in home care, then employed an auditory-guided neurofeedback-based strategy to modulate neural firing rates to select letters and to form words and sentences using custom software. Before implantation, this person was unable to express his needs and wishes through non-invasive methods, including eye-tracking, visual categorization of eye-movements, or an eye movement-based BCI-system. The patient started using the intracortical BCI system for voluntary verbal communication three months after implantation. With ALS progression, the patient lost the ability to open his eyes voluntarily as well as visual acuity, but he is still employing the auditory-guided neurofeedback-based strategy with his eyes closed to select letters and form words and sentences. Therefore, a CLIS patient who was unable to express his wishes and desires is employing the BCI system to express himself independent of vision.

## Results

One day after the implantation, attempts were initiated to establish communication. The patient was asked to use his previously effective communication strategy employing eye movements to respond to questions with known "yes" and "no" answers, which did not result in a classifiable neural signal, no difference in spike rate and multi-unit-activity (MUA). Passive movements of the patient's right fingers, thumb, and wrist evoked consistent neural firing rate modulations on several electrodes on both arrays. However, when we instructed the patient to attempt or imagine hand, tongue, or foot movements, we could not detect consistent responses. Subsequently, the communication strategy was changed on the 86th day after implantation, and neurofeedback-based paradigms (described in the Online Methods section) were employed, as shown in Fig. 1. In this setting, the patient was provided auditory feedback of neural activity by mapping a spike rate metric (SRM) for one or more channels to the frequency of an auditory feedback tone, as displayed in Fig. 1 (described in the "Neurofeedback communication" section of Online Methods, see sample Supplementary Video V1). The patient was able to modulate the sound tone on his first attempt on day 86 and subsequently was able to successfully modulate the neural firing rate and match the frequency of the feedback to the target for the first time on day 98. Employing the neurofeedback strategy, the patient was able to modulate the neural firing rate and was able to use this method to select letters and to free spell from day 106 onwards. The Results reported here include data from days 106–462 after implantation. Three of the authors (UC, NB and AT) frequently traveled to the patient's home to perform communication sessions about every two weeks for 3 or 4 consecutive days until February 2020. Because of the COVID pandemic from March 2020 to June 2020, all the sessions were performed via secured remote access to the patient's laptop. During these sessions, the patient's wife performed locally all required hardware connections, and the experimenters, either UC or AT, controlled the software remotely. During the experimental period reported here, the authors UC, AT, and NB performed experimental sessions on 135 days. The patient was hospitalised due to unrelated adverse events between days 120 and 145, 163 and 172, and 212 and 223 after implantation, during which time no sessions were performed.

Each session day, we started with a 10-minutes baseline recording, where the patient was instructed to rest. During this time period the experimenter ran a software program to determine the firing rate of different individual channels and select their parameters for the first neurofeedback session-block. Two different types of neurofeedback sessions were performed consecutively on each day, "feedback without reward" and "feedback with reward" with the goal (1) to select channels suitable for voluntary control by the patient and (2) to train the patient to control the selected channels' spiking activity voluntarily. The first paradigm ("feedback without reward") provided successive target tones, and the patient was asked to match the frequency of the feedback tone to the target tone. The second paradigm ("feedback with reward") was the same. However, upon reaching and holding (during a configurable number of interactions, each interaction lasting 250 ms) the feedback tone within a predefined range around the target frequency, an additional reward sound was delivered for 250 ms, indicating successful performance to the patient. Holding the feedback tone at the high (low) end of the range for a minimum of 250 ms was then interpreted as a successful "yes" ("no") response. After the first "feedback without reward" session, individual channels' firing rate distributions were automatically calculated. The experimenter selected channels with differential modulation for the high and low target tones and

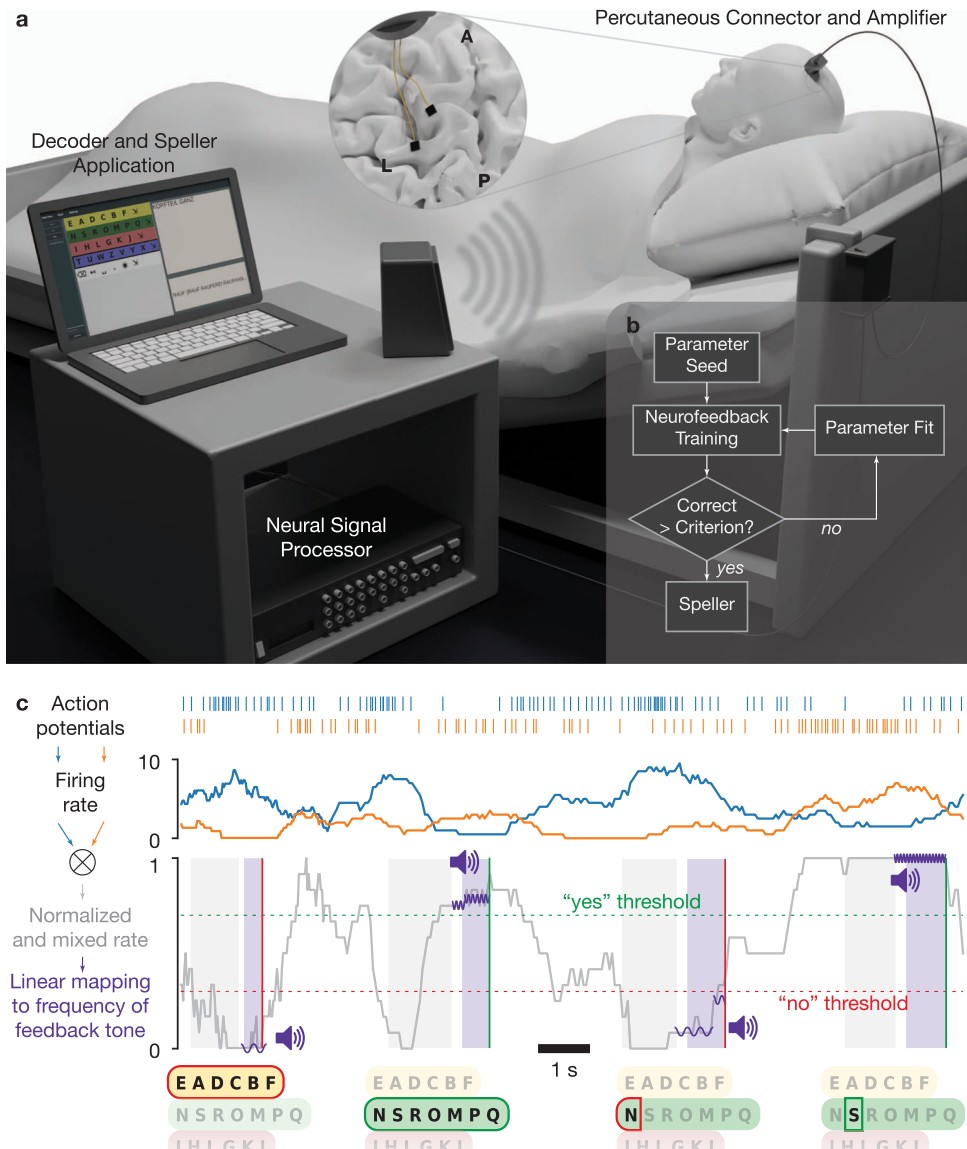

**Fig. 1 Setup and neurofeedback paradigm. a** Experimental setup. Two microelectrode arrays were placed in the precentral gyrus and superior frontal gyrus (insert, L: left central sulcus, A-P: midline from anterior to posterior). An amplifying and digitizing headstage recorded signals through a percutaneous pedestal connector. Neural signals were pre-processed on a Neural Signal Processor and further processed and decoded on a laptop computer. **b** Daily sessions began with Neurofeedback training. If the performance criterion was reached, the patient proceeded to speller use. If the criterion was not reached, parameters were re-estimated on neurofeedback data, and further training was performed. **c** Schematic representation of auditory neurofeedback and speller. Action potentials were detected and used to estimate neural firing rates. One or several channels were selected, their firing rates normalized and mixed (two channels shown here for illustration; see Online Methods). Options such as letter groups and letters were presented by a synthesized voice, followed by a response period during which the patient was asked to modulate the normalized and mixed firing rate up for a positive response and down for a negative response. The normalized rate was linearly mapped to the frequency of short tones that were played during the response period to give feedback to the patient. The patient had to hold the firing rate above (below) a certain threshold for typically 500 ms to evoke a "Yes" ("No") response. Control over the neural firing rates was trained in neurofeedback blocks, in which the patient was instructed to match the frequency of target tones.

updated the parameters for subsequent sessions. Employing this iterative procedure on each day, we performed several neuro-feedback blocks within a particular day to remind the patient of the correct strategy to control the firing rate, each typically consisting of 10 high-frequency target tones and 10 low-frequency target tones presented in pseudo-random order and also to tune and validate the classifier. Typically, if the patient could match the frequency of the feedback to the target in 80% of the trials, we proceeded with the speller.

**Neurofeedback sessions**. Figure 2a shows individual neurofeed-back trials, including an error trial, of one representative block. Over the reported period, there were 1176 feedback sessions as shown in Supplementary Fig. S1. In the 281 neurofeedback blocks preceding the speller blocks, 4936 of 5700 trials (86.6%) were correct (Fig. 2b), i.e., for target tone up (higher frequency) the decision was up (a "yes" answer), and for target down (low frequency) the decision was down (a "no" answer). The difference in error rates between 'up' and 'down' trials, i.e., the fraction of trials

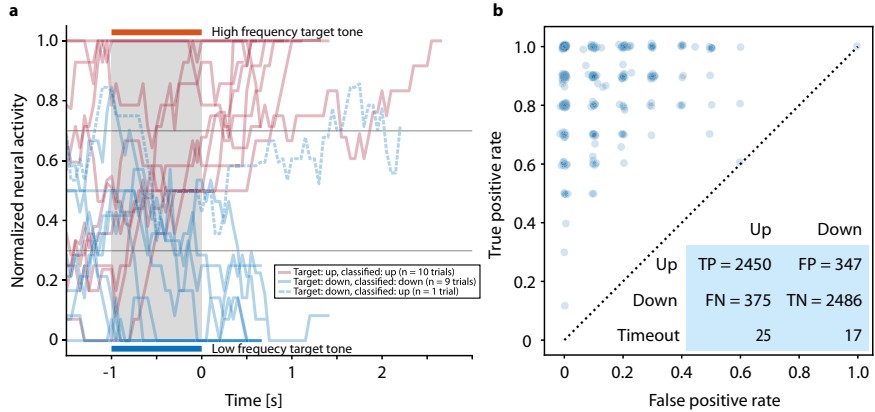

**Fig. 2 Neurofeedback task and classification. a** Representative example for normalized and mixed firing rate during ten high (red) and ten low (blue) target tone frequency feedback trials of day 247. The patient was asked to match the target tone by modulating the normalized and weighted firing rate, and he succeeded in all but one trial of this example. Trials were completed as soon as the firing rate was held above or below the upper or lower threshold, respectively. As defined in the Materials and Methods section, these feedback blocks were performed every day of recording for training, parameter selection, and validation of the selected parameters. The grey-shaded region from −1 to 0 s depicts the time period during which the high or low target tones were presented to the patient. The horizontal line at 0.3 and 0.7 shows the lower and upper threshold, respectively. Source data are provided as a Source Data file. **b** True positive rate vs. false positive rate of the trials in auditory neurofeedback blocks directly preceding speller blocks on days 123–462. Each circle represents one neurofeedback block; circles are jittered for better visibility. The blue insert at the bottom right corner shows the contingency table of all neurofeedback trials directly preceding speller blocks on days 123–462. In the blue insert—TP stands for true positive, i.e., up trials classified as up; FP stands for false positive, i.e, up trials classified as down; FN—stands for false negative, i.e., down trials classified as up; TN—stands for true negative, i.e., down trials classified as down; Time out denotes the trials that were unclassified. Source data are provided as a Source Data file.

in which the modulated tone did not match the target tone, (13.2% and 12.2%, respectively), was significant (Pearson's $\chi^2$ test: $p < 0.01$). The patient maintained a high level of accuracy in the neurofeedback condition throughout the reported period: on 52.6% of the days, the accuracy was at least 90% during at least one of the feedback trials blocks, i.e., the patient was able to match the frequency of the feedback to the target 18 out of 20 times. We observed considerable within-day variability of neural firing rates and hence performance of the neurofeedback classifier, necessitating manual recalibration throughout the day (see Supplementary Fig. S1). In the last feedback sessions before speller sessions, the median accuracy was 90.0%, the minimum was 50.0% (chance level). In 17.1% of the sessions, accuracy was below 80.0%.

**Speller sessions**. We continued with the speller paradigm when the patient's performance in a neurofeedback block exceeded an acceptance threshold (usually 80%). To verify that good performance in the neurofeedback task translated to volitional speller control (based on correct word spelling), we asked the patient to copy words before allowing free spelling. On the first three days of speller use, the patient correctly spelled his own, his son's, and his wife's names. After an unrelated stay at the hospital, we again attempted the speller using the same strategy on day 148.

Afterward, we relied on a good performance in the neurofeedback task, i.e., the patient's ability to match the frequency of the feedback to the target in 80% of trials, to advance to free spelling. The selection of two letters from a speller block on day 108 is shown in Fig. 3. Supplementary Video V2 presents a representative speller block.

Over the reported period, out of 135 days, speller sessions were attempted on 107 days, while on the remaining 28 days use of the speller was not attempted because the neurofeedback performance criterion was not reached. The patient produced intelligible output, as rated independently by three observers, on 44 of 107 days when the speller was used (Fig. 4). On average, 121 min were spent spelling and the average length of these

communications was 131 characters per day. The patient's intelligible messages comprised 5747 characters produced over 5338 min, corresponding to an average rate of 1.08 characters per minute. This rate varied across blocks (min/median/max: 0.2/1.1/ 5.1 characters per minute). Over the reported period, there was no apparent trend in spelling speed. There were 312 pairs of speller blocks and preceding neurofeedback blocks. The speller output was rated 0 for unintelligible by raters, 1 for partially intelligible, 2 for intelligible. The Spearman correlation between Neurofeedback task accuracy and subsequent speller intelligibility was 0.282 ($p = 4.002\mathrm{e}{-07}$). The Spearman correlation between Neurofeedback accuracy and number of letters spelled was 0.151 ($p = 7.671\mathrm{e}{-03}$). The information transfer rate (ITR) during intelligible speller sessions was 5.2 bits/minute on average (min/ median/max: 0.3/4.9/21.4 bits/minute).

On the second day of free spelling, i.e., on the 107th day after implantation, the patient spelled phrases, spelled in three-time episodes, thanking NB and his team ('erst mal moechte ich mich niels und seine birbaumer bedanken' – 'first I would like to thank Niels and his birbaumer'). Many of the patient's communications concerned his care (e.g. 'kop?f immerlqz gerad' – 'head always straight', day 161; 'kein shirt aber socken' – 'no shirts but socks [for the night]', day 244; 'mama kopfmassage' – 'Mom head massage', day 247; 'erstmal kopfteil viel viel hoeh ab jetzt imm' – 'first of all head position very high from now', day 251; 'an alle muessen mir viel oefter gel augengel' – 'everybody must use gel on my eye more often', day 254; 'alle sollen meine haende direkten auf baubch' – 'everybody should put my hand direct on my stomach', day 344; 'zum glotze und wenn besuchen da ist das kopfteil immer gaaanz rauf' – 'when visitors are here, head position always very high' on day 461. The patient also participated in social interactions and asked for entertainment ('come tonight [to continue with the speller]', day 203, 247, 251, 294, 295, 'wili ch tool balbum mal laut hoerenzn' – 'I would like to listen to the album by Tool [a band] loud', day 245, 'und jetwzt ein bier' – 'and now a beer', day 247 (fluids have to be inserted through the gastro-tube), 251, 253, 461. He even gave suggestions to improve his speller performance by spelling 'turn on word

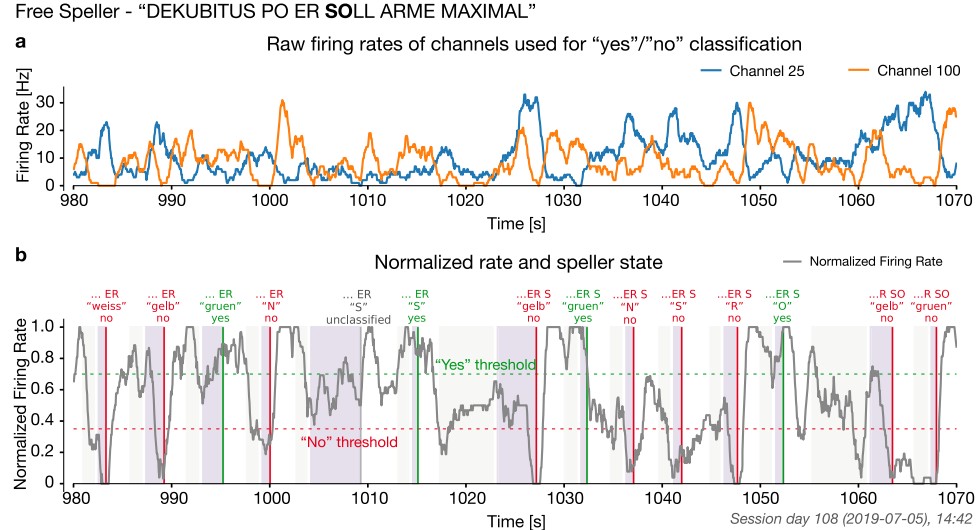

**Fig. 3 Example of letter selection during a free spelling block. a** Firing rate of the channels 25 and 100 used for "yes"/"no" classification on day 108. **b** Normalized firing rate and the speller state during the same 90 s period of a speller block. "Yes"/"no"/ timeout decisions are marked by vertical lines and the option selected in green and not selected in red. This example is part of the phrase "dekubitus po er soll arme maximal", referring to bedsore and instructing the aide to change arm position. Source data are provided as a Source Data file.

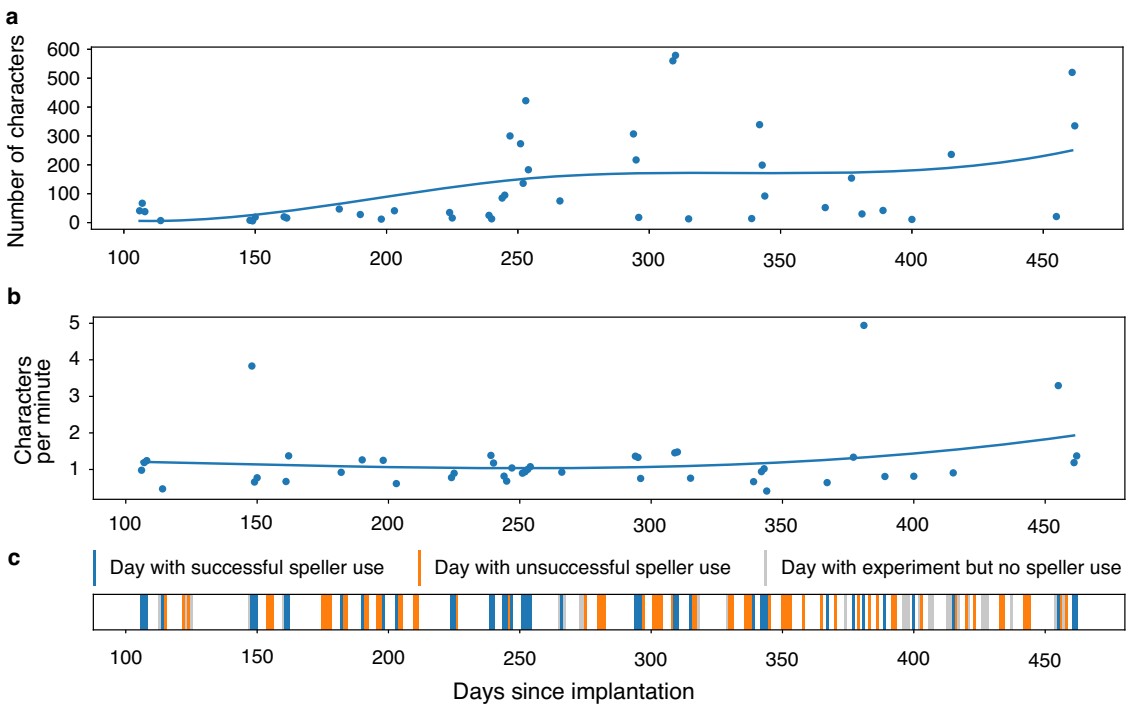

**Fig. 4 Overview of BCI use. a** Number of characters spelled by the patient during speller sessions whose output was rated 'intelligible' (rating described in text), aggregated by day. **b** Characters selected per minute in 'intelligible' speller sessions, aggregated by day. **c** Speller use during the period presented. Sessions span 135 days. Green bars represent days on which speller was used and yielded intelligible output (44 days). Yellow bars represent days on which speller use was attempted, but no intelligible output was produced (63 days). On 28 days, speller use was not attempted (red). Source data are provided as a Source Data file.

recognition' on day 183, 'is it easy back once confirmation' on Day 253, 'tell alessandro i need to save edit and delete whole phrases and all of that into the list where (patient's son name)' on day 295, 'why cant you leave the system on. ifind that good' on day 461, in English as the patient knew that the experimenter UC and AT are not native German speakers and mostly spoke in English with the patient. On day 247 he gave his feedback on speller as, 'jungs es funktioniert gerade so muehelos', - 'boys, it

works so effortlessly'. The patient expressed his desire to have different kind of food in his tube as, 'mixer fuer suppen mit fleisch' – 'instructed his wife to buy a mixer for soup with meat' on day 247'; 'gulaschsuppe und dann erbsensuppe' – 'Gulash soup and sweet pea soup' on day 253; 'wegen essen da wird ich erst mal des curry mit kartoffeln haben und dann bologna und dann gefuellte und dann kartoffeln suppe' – 'for food I want to have curry with potato then Bolognese and potato soup on day 462. He

interacted with his 4 years old son and wife, '(son's name) ich liebe meinen coolen (son's name) – 'I love my cool son' on day 251; '(son's name) willst du mit mir bald disneys robin hood anschauen'- 'Do you want to watch Disney's Robin Hood with me' on day 253; 'alles von den dino ryders und brax autobahnund alle aufziehautos' – 'everything from dino riders and brax and cars' on day 309; '(son's name) moechtest du mit mir disneys die hexe und der zauberer anschauen auf amazon' – 'would you like to watch Disney's witch and wizard with me on amazon' on day 461; 'mein groesster wunsch ist eine neue bett und das ich morgen mitkommen darf zum grillen' – 'My biggest wish is a new bed and that tomorrow I come with you for barbecue' on day 462.

## Discussion

We demonstrate that a paralyzed patient, according to the presently available physiological and clinical criteria in the completely locked-in state (CLIS), could volitionally select letters to form words and phrases to express his desires and experiences using a neurally-based auditory neurofeedback system independent of his vision. The patient used this intracortical BCI based on voluntarily modulated neural spiking from the motor cortex to spell semantically correct and personally useful phrases. Properties of the multielectrode array impedance and recordings across sessions are shown in the Supplementary Fig. S2. In all blocks, measurable spike rate differentiation between "yes" and "no" during the neurofeedback trials and "select" and "no select during speller blocks appeared in only a few channels in the SMA (supplementary motor area) out of all active channels, as shown in Supplementary Fig. S3a. After the establishment of successful communication after day 86, similar channels from the Supplementary Motor Cortex array were used for communication sessions with the patient, as shown in Supplementary Fig. S3b. Mainly electrode 21 and neighbouring electrodes were used, which demonstrated differential control of the feedback tones during the neurofeedback sessions before spelling. Because the neurofeedback procedure was the prerequisite for successful communication after 86 days of attempted, but unsuccessful decoding, a multichannel decoding algorithm was not implemented following our clinical judgment based on learning principles[23] that such a substantial change in the procedure might impede or extinguish the successful control of the patient and spelling. In addition, after this failure, we attempted a neural feedback approach, based on learning principles, capitalizing on the observation that neural firing rates could be used to achieve levels suitable to make yes/no choices. For the speller sessions, only one to four channels were used for control, as shown in Supplementary Fig. S3. There was insufficient time to explore other decoding approaches, and we wanted to establish that communication was feasible at all in CLIS. We cannot explain why the other electrodes did not provide modulation suitable for multichannel decoding. Perhaps with further sessions and other strategies, not possible in this experiment, we might have identified faster, or more accurate approaches. Speller use duration was highly variable, ranging from a few minutes to hours. As shown in Fig. 4, the patient generated a different number of characters on different days. He spelled only under 100 characters on some days, while on other days, he produced more than 400 characters. Despite the huge variation in the number of characters spelled, the number of characters spelled per minute was mostly around 1 character per minute, and ITR averaged 5 bits/minute. Communication rates are lower than in other studies using intracortical arrays[7–9], but comparable to EEG P300 spellers for ALS patients[24,25] and much faster than an SSVEP EEG BCI for advanced ALS patients[12]. These apparent poor performances are primarily due to the completely auditory nature of these systems,

which are intrinsically slower than a system based on visual feedback. Lastly it was noteworthy that free voluntary spelling mainly concerned requests related to body position, health status, food, personal care and social activities suggesting that even with this slow speller the patient could relay his needs and desires to caretakers and family.

Our study showed communication in a patient with CLIS. It is worth mentioning that no universally accepted clinical definition exists to distinguish LIS from CLIS; the current standard criteria to differentiate LIS from CLIS is the presence or absence of means of communication. During the transition from LIS to CLIS, patients are initially left with limited, and finally, no means of communication. The time course of this transition process is patient and disease specific. In theory, other voluntary muscles than eye-movements could have been used for Electromyography (EMG)-based communication attempts. Particularly face muscles outside the extraocular muscles may remain under voluntary control in some cases even after the loss of eye-muscle control. To the best of the authors' knowledge, no study has extensively investigated the remaining muscle activity of CLIS patients, but in previous studies[22,26] the authors showed that during the transition from LIS to CLIS some remaining muscles of the eyes continue to function and can be used for successful communication.

In the case of the patient described here, extensive electro-oculogram (EOG) recordings were performed to demonstrate that no other measurable neuromuscular output existed- a way to confirm CLIS. The patient employed an EOG-based BCI for communication successfully for the last time in February 2019 when the amplitude of EOG signal decreased below 20 μV. Nevertheless, extensive post-hoc EOG analysis showed a significant difference in the maximum, mean, and variance feature of the eye movement corresponding to "yes" and "no" even after the patient's inability to employ the EOG-based system. This failure to communicate despite the presence of a significant difference in some of the features may be due to the limitations of the EOG-based BCI system. However, as differences of eye movement amplitudes were only detectable over tens of trials and not reliably from session to session, EOG was not a practical signal for communication. In this study, caretakers and family members denied the existence of any possible reliable communication from February 2019 onwards, when this study occurred. Thus, we conclude based on our reported measurements that the patient described was in a CLIS a few weeks before and also after implantation. This statement does not exclude the possibility that even more sensitive measurements of somatic-motor control could reveal some form of volitional control, which would render the diagnostic statement of CLIS at least for this case inaccurate. Nevertheless, by the measures we describe here no muscle-based signals useful for communication were evident, leading us to conclude that this patient could be classified as in the chronic complete locked in state but was able to communicate using an implanted BCI system usefully.

The present BCI communication demonstrates that an individual unable to move for protracted periods is capable of meaningful communication. Still, the current neurofeedback based BCI system has several limitations, as several software and hardware modifications would need to be implemented before the system could be used independently by the family or caretakers without technical oversight. The BCI-software is presently being modified to improve communication quality and rate and the self-reliance of the family.

In this study, communication rates were much lower compared to other studies using intracortical arrays, which include communication with a point-and-click screen keyboard[7–9], and decoding of imagined handwriting in a spinal cord injured patient[27]. People with ALS and not apparently completely locked-

in have been able to use multi neuron-based decoders for more rapid communication than seen here. The differences might be both technical—a failure of the electrode—or biological, i.e., related to the disease state. While the multielectrode array (MEA) used here typically shows some variable level of degradation in the quality and number of recordings over time, such arrays reportedly provide useful signals for years[28,29].

Our MEA retained impedances in the useful range across the entire experimental period of this study (Fig. S2) and neurons were recorded on many channels suggesting that the loss of recordable neural waveforms cannot explain performance differences. The most striking difference in the present data was the inability to record neurons that modulated with the participant's volitional intent. This could be the result of the disease processes on the neurons themselves or the protracted loss of sensory-motor input itself. The observation that control was intermittent may reflect changes in neural connectivity or the ability to be activated. Lack of any somatic sensory feedback, especially that from muscles, might impede voluntary modulation of neural activity. Participants with ALS enrolled in previous trials apparently had at least some residual voluntary control of muscles, whereas this participant had lost all control by the time of implantation. Additionally, advanced ALS may have led to cognitive or affective changes such as shortened attention span or modified motivational systems that may have made it difficult to achieve reliable modulation of large numbers of neurons (dozens to hundreds) achieved in other ALS participants with similar BCI systems implanted. Altered cortical evoked response amplitudes and latencies[30] seen in this individual may be a reflection of these abnormal states. CLIS patients with ALS show highly variable and often pathological neurophysiological signatures[31] such as heterogeneous sleep-waking cycles[32] that may also affect the ability to engage neurons. Lastly, auditory cues may engage motor processes that will activate neurons in frontal areas outside of the motor cortex[33], which may have contributed to the changes with the auditory task used here.

To conclude, this case study has demonstrated that a patient without any stable and reliable means of eye-movement control or identifiable communication route employed a neurofeedback strategy to modulate the firing rates of neurons in a paradigm allowing him to select letters to form words and sentences to express his desires and experiences. It will be valuable to extend this study to other people with advanced ALS to address the aforementioned issues systematically.

## Methods

The medical procedure was approved by the Bundesinstitut für Arzneimittel und Medizinprodukte ("BfArM", The German Federal Institute for Drugs and Medical Devices). The study was declared as a Single Case Study and has received a special authorization ("Sonderzulassung") by BfArM, according to §11 of the German Medical Device Law ("Medizin-Produkte-Gesetz") on December 20, 2018, with Case Nr. 5640-S-036/18. The Ethical Committee of the Medical Faculty of the Technische Universität München Rechts der Isar provided support to the study on 19 Jan 2019, along with the explicit permission to publish on 17 February 2020. Before the patient transitioned into CLIS, he gave informed consent to the surgical procedure using his eye movements for confirmation. The patient was visited at home by authors HT and JL, and thorough discussions were held with the legally responsible family members (wife and sister) in order to establish convincing evidence of the patient's informed consent and firm wish to undergo the procedure. The legally responsible family members then provided informed written permission to the implantation and the use of photographs, videos, and portions of his protected health information to be published for scientific and educational purposes. In addition, a family judge at the Ebersberg county court gave the permission to proceed with the implantation after reviewing the documented consent and a visit to the patient. The patient received no compensation for the participation.

**Patient**. The patient, born in 1985, was diagnosed with progressive muscle atrophy, a clinical variant of non-bulbar ALS, selectively affecting spinal motor neurons in August 2015. He lost verbal communication and the capability to walk by the end of 2015. He has been fed through a percutaneous endoscopic gastrostomy tube and

artificially ventilated since July 2016 and is in home care. He started using the MyTobii eye-tracking-based assistive and augmentative communication (AAC) device in August 2016. From August 2017 onwards, he could not use the eye-tracker for communication because of his inability to fixate his gaze. Subsequently, the family developed their own paper-based spelling system to communicate with the patient by observing the individual's eye movements. According to their scheme, any visible eye movement was identified as a "yes" response, lack of eye movement as "no". The patient anticipated complete loss of eye control and asked for an alternative communication system, which motivated the family to contact authors NB and UC for alternative approaches. Initial assessment sessions were performed in February 2018. During this interval, the detection of eye movements by relatives became increasingly difficult, and errors made communication attempts impossible up to the point when communication attempts were abandoned. The patient and family were informed that a BCI-system based on electrooculogram (EOG) and/or electroencephalogram (EEG) might allow "yes" - "no" communication for a limited period.

The patient began to use the non-invasive eye movement-based BCI-system described in Tonin et al.[21]. The Patient was instructed to move the eyes ("eye-movement") to say "yes" and not to move the eyes ("no eye-movement") to say "no". Features of the EOG signal corresponding to "eye-movement" and "no eye-movement" or "yes" and "no" were extracted to train a binary support vector machine (SVM) to identify "yes" and "no" response. This "yes" and "no" response was then used by the patient to auditorily select letters to form words and hence sentences. The patient and family were also informed that non-invasive BCI-systems might stop functioning satisfactorily, and in particular, selection of letters might not be possible if he became completely locked-in (where no eye movements could be recorded reliably). In that case, implantation of an intracortical BCI-system using neural spike-based recordings might allow for voluntary communication. As the patient's ability to communicate via non-invasive BCI systems deteriorated, in June 2018, preparations for the implantation of an intracortical BCI system were initiated. To this end, HT and JL and GF were approached in order to prepare the surgical procedure and provide clinical care in a hospital close to the patient's home. The patient was able to use the non-invasive BCI system employing eye-movement to select letters, words, and sentences until February 2019, as described in Tonin et al.[21,34]. By the time of implantation, the EOG/EEG based BCI system failed, as signals could not be used reliably for any form of communication in this investigational setting. The EOG/EEG recordings and their analysis are described in Supplementary Note 1 and Supplementary Fig. S4. Additionally, the patient reported low visual acuity caused by the drying of the cornea.

**Surgical procedure**. A head MRI scan was performed to aid surgical planning for electrode array placement. The MRI scan did not reveal any significant structural abnormalities, in particular no brain atrophy or signs of neural degeneration. A neuronavigation system (Brainlab, Munich, Germany) was used to plan and perform the surgery. In March 2019, two microelectrode arrays (8×8 electrodes each, 1.5 mm length, 0.4 mm electrode pitch; Blackrock Microsystems LLC) were implanted in the dominant left motor cortex under general anaesthesia. After a left central and precentral trephination, the implantation sites were identified by neuronavigation and anatomical landmarks of the brain surface. A pneumatic inserter was used[28] to insert the electrode arrays through the arachnoid mater, where there were no major blood vessels. The pedestal connected to the microelectrode arrays connected via a bundle of fine wires (Blackrock Microsystems LLC), was attached to the calvaria using bone screws and was exited through the skin. The first array was inserted into the hand area region of the primary motor cortex[35], and the second array was placed 2 cm anteromedially from the first array into the region of the supplementary motor area (SMA) as anatomically identified. No implant-related medical adverse events were observed. After three days of postoperative recovery, the patient was discharged to his home.

**Neural signal processing**. A digitizing headstage and a Neural Signal Processor (CerePlex E and NSP, Blackrock Microsystems LLC) were used to record and process neural signals. Raw signals sampled at 30kS/s per channel were bandpass filtered with a window of 250–7500 Hz. Single and multi-unit action potentials were extracted from each channel by identifying threshold crossings (4.5 times root-mean-square of each channel's values). Depending on the activity and noise level, thresholds were manually adjusted for those channels used in the BCI sessions after visual inspection of the data to exclude noise but capture all of the visible spikes above the threshold. Neural data were further processed on a separate computer using a modified version of the CereLink library (https://github.com/dashesy/CereLink) and additional custom software. For communication, we used spike rates from one or more channels. A spike rate metric (SRM) was calculated for each channel by counting threshold crossings in 50 ms bins. The SRM was calculated as the mean of these bins over the past one second.

Custom software written in Python and C + + was used to perform and control all BCI sessions. The software managed the complete data flow of the raw signals provided by the NSP, allowing manual configuration of recording parameters, selection of individual channels for neurofeedback and storing of neural data, and meta-information (timing information of trigger events, etc.) required for offline analysis.

The software enabled the experimenter to configure different experimental protocols, to select an experimental paradigm for each session, and to trigger the start and end of a session. The software-controlled the presentation of auditory stimuli to the patient, including the presentation of feedback from his neural activity. It also provided live feedback to the experimenter regarding ongoing progress, e.g., the currently spelled phrase. Also, the software provided live visualization of neural activity, including the original firing rate activity of selected channels and normalized firing rate activity used for neurofeedback. To secure smooth real-time processing and to avoid potential performance bottlenecks, the software supported multiprocessing. That is, all critical processes, including data acquisition, data storage, neurofeedback, classification, and visualization, were executed in separated cores.

**Neurofeedback communication**. The patient was provided auditory feedback of neural activity levels by mapping the SRM for one or more channels to the frequency of an auditory feedback tone, as shown in Fig. 1. Single channel spike rates were normalized according to the spike rate distribution of each channel. Selected channels' normalized SRMs were then summed and linearly mapped to the range of 120–480 Hz, determining the frequency of the feedback tone produced by an audio speaker. Feedback tones were updated every 250 ms. The firing rate $r_i$ of each selected channel was constrained to the range $[a_i, b_i]$, normalized to the interval $[0,1]$, and optionally inverted, and the resulting rates were averaged:

$$r(t) = \frac{1}{n} \sum_i^n \frac{1 - c_i}{2} + c_i \frac{\max(\min(r_i(t), b_i), a_i)}{b_i - a_i} \tag{1}$$

where $r(t)$ is the overall normalized firing rate, and the $c_i$ are 1 or $-1$. The normalized rate was then linearly mapped to a frequency between 120 and 480 Hz for auditory feedback. Feedback tones were pure sine waves lasting 250 ms each. Initially, channels were selected randomly for feedback. Then the parameters $a_i, b_i, c_i$ as well as the channels used for control were chosen and iteratively optimized each day in the neurofeedback training paradigms.

The first paradigm ("feedback without reward") provided successive target tones at 120 or 480 Hz, and the patient was asked to match the frequency of the feedback to the target (typically 20 pseudorandom trials per block). In the "feedback with reward" paradigm, was essentially the same, however, upon reaching and holding (during a configurable number of interactions, each interaction lasting 250 ms) the feedback tone within a predefined range around the target frequency, an additional reward sound was delivered for 250 ms indicating successful performance to the patient. Holding the feedback tone at the high (low) end of the range for 250 ms was then interpreted by the patient upon instruction as a Yes (No) response (see Supplementary Video V1 as a typical example). The "feedback with reward" paradigm served to train and validate the responses.

We also validated the Yes/No responses in a question paradigm, in which the answers were assumed to be known to the patient. Furthermore, we used an 'exploration' paradigm to test if the patient's attempted or imagined movements could lead to modulation of firing rates.

Finally, in an auditory speller paradigm, the patient could select letters and words using the previously trained Yes/No approach. The auditory speller paradigm is depicted in Fig. 1c. The speller system described here avoids long adaptation and learning phases because it is identical to the one used previously when he was still in control of eye movements. The original arrangement of the letters in their respective groups was chosen according to their respective frequency in the patient's native German language.

The speller's output was rated for intelligibility by three of the authors (UC, IV, and JZ). Three categories were used: unintelligible, ambiguous, and intelligible. Ambiguous speller output includes grammatically correct words that could not be interpreted in the context as well as strings of letters that could give rise to uncertain interpretations. Intelligible phrases may contain words with spelling mistakes or incomplete words, but the family or experimenter identified and agreed upon their meaning.

To evaluate the performance of the speller, the information transfer rate[36] (ITR) $B$ during speller sessions that were rated as intelligible was calculated as:

$$B = \log_2 N + P \log_2 P + (1 - P) \log_2 \frac{1 - P}{N - 1} \tag{2}$$

where $N$ is the number of possible speller selections (30 including space, delete, question mark and end program), and $P$ is the probability that a correct letter was selected. Multiplication with selected symbols and division by session duration yields bits per minute.

**Data handling**. Software and procedures were designed to provide redundancy and automation to ensure that crucial information is always saved with each recording:

1. The BCI software was implemented with extensive automated logging for each session:

   a. neural data (spike rates) used for BCI control
   b. event timestamps from the neurofeedback training/validation and speller paradigms

   c. configuration used to run the particular session, including channel selection, normalization parameters, thresholds for yes/no detection, task timings, arrangement of letters in speller, etc.
   d. source code of the KIAP BCI software used on that day. The BCI software was kept under version control using git. The hash of the current commit was saved along with any changes compared to that commit.

2. Specific instructions were given to the personnel performing the experimental sessions to acquire raw neural data collected in parallel to the BCI data, which included loading a configuration file, starting data recording before a BCI session, and stopping the recording at the end. Two experimenters were on-site, when possible, to divide system operation and patient interaction tasks.

3. Information about each recording session was entered into a session log in a shared Excel file (which has a history of edits). Information logged include for each session:

   a. kind of experiment
   b. file names of raw data and KIAP BCI data
   c. any additional EEG recordings if performed
   d. names of video files if performed
   e. experimenters present
   f. observations/abnormalities for the session
   g. data recording abnormalities, etc.

4. During the experiments, known issues were fixed, for example, a change in log file format was implemented (as noted in the accompanying dataset), which allowed to more easily interpret the data. The post hoc analysis, i.e., parsing log files and data compilation was checked manually for several sessions. Co-authors reviewed the results and the process.

5. Data handling procedures were implemented to ensure that data integrity was maintained from recording to safe storage.

**Dataset reported in this article**. The dataset here spans days 106 to 462 after implantation. For the analysis of neurofeedback trials in Fig. 2 and the corresponding main text, only blocks after day 123 were used because of a change in paradigm (before day 123, incorrect trials and time-outs were not differentiated). For Supplementary Fig. 1, all neurofeedback blocks were used, as time-out trials were counted as 'incorrect' as well. All speller sessions performed between days 106 and 462 were included in the analysis. The BCI data of one neurofeedback and one speller session were lost during data transfer and the loss was only discovered after the original data had been deleted. These sessions were therefore excluded from the analysis.

**Reporting summary**. Further information on research design is available in the Nature Research Reporting Summary linked to this article.

## Data availability

The data upon which the findings in this paper are based (neural firing rates, event log files for data presented in Figs. 2, 3, 4, S1, S3; electrode impedances and spike event files for data presented in Fig. S2) are available at https://doi.org/10.12751/g-node.jdwmjd[37]. The EOG data which Fig. S4 is based on is available at https://doi.org/10.12751/g-node.ng4dfr[38]. Source data are provided with this paper. The raw neural recordings is available upon request to J.B.Z, yet owing to the potential sensitivity of the data, an agreement between the researcher's institution and the Wyss Center is required to facilitate the sharing of these datasets. Source data are provided with this paper.

## Code availability

The code used to run the BCI system is available at https://doi.org/10.12751/g-node.ihc6qn[39].

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

## Acknowledgements

This research was supported by the Wyss Center for Bio and Neuroengineering, Geneva, Deutsche Forschungsgemeinschaft (DFG BI 195/77-1) – N.B. and U.C.; German Ministry of Education and Research (BMBF) 16SV7701, CoMiCon – N.B. and U.C.; LUMINOUS-H2020-FETOPEN-2014-2015-RIA (686764) – N.B. and U.C.; Bogenhausen Staedtische Klinik, Munich. The authors thank Andrew Jackson and Nick Ramsey for their comments on an earlier version of the manuscript. Aleksander Sobolewski contributed the 3D model for Fig. 1. We thank the patient and his family.

## Author contributions

U.C.—Initiation; Conceptualization; Ethics Approval; Performed 95% of the sessions with the patient before and after implantation; Figures; Neurofeedback paradigm; Manuscript writing. I.V.—Software development, integration and testing; Data analysis; Neurofeedback paradigm implementation; Performed 5% of the sessions after the implantation. J.B.Z.—Neurofeedback paradigm implementation; Data analysis; Figures; Manuscript writing. A.E.—Software testing; EEG/EOG analysis; Figures. A.T.—Speller software development; Performed 30% of sessions before implantation and 20% of the sessions after implantation; EEG/EOG analysis; Figures. A.J.-G.—EEG/EOG analysis; Figures. M.K.A.—Graphical user interface. H.R.T.—Ethics Approval; Medical patient care; Clinical and diagnostic neurological procedures. J.L.—Ethics approval; Neurosurgery; Clinical care. G.M.F.—Neurosurgical training. A.W.—Ethics approval; BfArM approval; Neurosurgical training. J.P.D.—Initiation; Conceptualization, writing, review, editing. N.B.—Initiation; Conceptualization; Coordination, Clinical-psycholgical procedures and care; Neurofeedback paradigm; performed 30% of the sessions with UC; Ethics approval; BfArM approval; Manuscript writing.

## Competing interests

The authors declare no competing interests.
