## [Peer Review File · Nature Communications]

Spelling Interface using Intracortical Signals in a Completely Locked-In Patient enabled via Auditory Neurofeedback TrainingREVIEWER COMMENTS

Reviewer #1 (Remarks to the Author):

This is an excellent and important study, which will be read with strong interest by the community. Given the difficult nature of this study, and the lack of any previous work with multichannel invasive devices with this important population in the literature, the limitation of having only one subject is to be expected, and should not preclude publication.

However, there is additional information that should be provided from the experiments that have already been done. First of all, how does this person communicate at home? Much effort and data was devoted to showing diminishing EOG for example, but it was implied at several times that the family is still able to get some information from the participant outside of these studies, during 2019. Also, what is the state of this Utah array and these experiments as of the present submission date? With $n=1$, the completeness of this dataset up to the time of publication is critical, even if further data is also reserved for future papers.

The statement of "one or more" neurons in the Methods should be specified precisely. There are few enough experimental days to list what was done every day, or else it would be informative to describe in detail what was most commonly done, and whether this relied on the same 1-2 channels throughout the year.

Similarly, only successful days are described in the Results when half of the days did not result in free spelling. The graphs show that there was successful training and yes/no questions asked on many of the days where this did not work. The two datasets should be analyzed together to say whether a successful training and question experiment indicated intelligible use, or not. More examples of speech should be given from non-successful days, and there should be more analysis of the neural data and training datasets on those days to look into whether failures can be explained.

Reviewer #2 (Remarks to the Author):

In this study, the authors demonstrate a system that enables volitional communication for an individual in a completely locked-in state, which has not been demonstrated before. Unlike systems developed for those in a locked-in state which rely on visual feedback, this system utilizes auditory feedback to assist the user in spelling words, thus eliminating the need for a functional ocular system to communicate.

The authors' conceptual approach to designing a communication system for a user that is completely locked-in makes sense as long as the ocular system cannot be used for feedback at all. It is unclear from the article, but if visual feedback could be presented without requiring eye movements, perhaps an approach like Vansteensel et al., 2016 would be preferable since the communication rates are higher. As it stands, the article also lacks sufficient analysis and quantification of data that led to design decisions of the system and performance outcomes of the system. With the inclusion additional analyses and justification of auditory feedback usage, this paper presents an advance in developing communication tools for people in a completely locked-in state. Please see my specific comments below.

Line 123: Did the authors attempt to establish communication in between days 1 and 86? I would imagine that the day after implantation would not be an accurate representation of the participant's brain state as the swelling would need to subside. Were the tones and reward signals played to participants before neurofeedback training? Did they elicit any neural response? It would be great to see plots of the tone related spiking rates before and after starting the neurofeedback task on day 86.

Furthermore overall brain mapping showing the SRMs for all channels should be shown. The authors rely on one or two channels for their system despite having 128 channels to work with. Many decoders with inputs from the motor cortex rely on many channels to achieve high

performance, why was a multi-channel approach not taken here? In general, it seems that the reader is left in the dark as to the characteristics of the neural responses (or lack thereof) that are observed corresponding to various behavior attempts and stimuli.

Line 123: Could the authors provide more detail on the lack of neural activations for attempted hand movements and contrast their findings with Freudenburg et al., 2019 (Frontiers in Neuroscience)?

Line 173: Could the authors provide the information transfer rate for their results? This will help establish a performance baseline for futures studies. Also, including the character error rate and word error rate when in spelling mode would be helpful.

Line 346: The authors use two forms of neurofeedback (tones and a reward sound), but there is no analysis or discussion on how these forms of feedback influenced the task. Are both forms of feedback necessary, or could the task be performed with the reward feedback alone? How important is linearizing the tone response as opposed to taking a more binary classification approach?

Supp. Figure 1: Plotting the mean/median performance instead of the maximum performance is preferable. Given the high performance variability, the maximums do not seem meaningful. On that note, why is the variance across trial sets so high? It may be beneficial to the read to see mean/std event related spiking rate plots of selected channels during the tone and spelling tasks. Would using a multi-channel system help stabilize the performance of the system?

Figure 1B and 4 captions do not adequately explain the figures. The caption should describe all aspects of figures.

Figure 2: The figure focuses primarily on duration of the BCI usage, but perhaps a better aspect of the data is to highlight the performance of the system on each day, especially the speller. Instead of using a subjective three-way classification for speller performance, an objective measure like character error rate should be used. This figure could be two panels where an adaptation of supp. Figure 1 is B.

Figure 3A. There is no indication of what the grey bar represents in the caption, I'm assuming it is the tone presentation period.

REVIEWER COMMENTS

Reviewer #1 (Remarks to the Author):

1. This is an excellent and important study, which will be read with strong interest by the community. Given the difficult nature of this study, and the lack of any previous work with multichannel invasive devices with this important population in the literature, the limitation of having only one subject is to be expected, and should not preclude publication.

Response: We would like to thank the reviewer for the supportive comment.

2. However, there is additional information that should be provided from the experiments that have already been done. First of all, how does this person communicate at home? Much effort and data was devoted to showing diminishing EOG for example, but it was implied at several times that the family is still able to get some information from the participant outside of these studies, during 2019. Also, what is the state of this Utah array and these experiments as of the present submission date? With $n=1$, the completeness of this dataset up to the time of publication is critical, even if further data is also reserved for future papers.

Response: We would like to thank the reviewer for the comment and state that the last time the patient's family could decipher the patient's eye-movement to "yes" and "no" questions was in February 2019, after which the patient family was unable to decipher his eye-movement anymore. Even though the patient was implanted in March 2019, we could not establish any communication with the patient for three months after implantation. Only after implementing the modulation of the sound-based neurofeedback strategy in July 2019, we were able to establish communication with the patient as described in this manuscript **from line 115 to 130**. However, from time to time, as mentioned in the paper we used the EOG based BCI system to assess whether the patient could still use his eye-movement for communication and as reported in the manuscript, these attempts always failed after Feb 2019 as his performance was random. In the revised manuscript we have updated the introduction section with the patient's communication ability before and after implantation, as shown below.

Changes in the manuscript: Introduction section – Line 115 to 130

“Here, we established that an individual was in the CLIS state and demonstrated that sentence-level communication is possible without relying upon the patient's vision. This individual lacked reliable voluntary eye-movement control and, consequently, was unable to use an eye-tracker for communication. The patient was also ultimately unable to use a non-invasive eye-movement-based computerized communication system²². To restore communication in CLIS, this participant was implanted with intracortical microelectrode arrays in two motor cortex areas. The patient, who is in home care, then employed an auditory-guided neurofeedback-based strategy to modulate neural firing rates to select letters and to form words and sentences using custom software. Before implantation, this person was unable to express his needs and wishes through non-invasive methods, including eye-tracking, visual categorization of eye-movements, or an eye movement-based BCI-system. The patient started using the intracortical BCI system for voluntary verbal communication three months after implantation. With ALS progression, the patient lost the ability to open his

eyes voluntarily as well as visual acuity, but he is still employing the auditory-guided neurofeedback-based strategy with his eyes closed, selects letters and forms words and sentences. Therefore, a patient who was unable to express his wishes and desires is employing the BCI system to express himself independent of vision.”

We agree with all the reviewer's comments on providing information about the Utah array status and status of this experiment and dataset. In the revised manuscript, we have included the data until the first date of submission, i.e., until 23rd June 2020, and updated the relevant figures with the extended data. Therefore **Figure 2**, **Figure 4** and **Supplementary Figure S1** have been updated with the further analysis of the extended dataset, as shown below.

Changes in the manuscript: Modified the Figure 2 with the extended dataset

Caption - Figure 2: Neurofeedback task and classification– a) Representative example for normalized and mixed firing rate during ten high (red) and ten low (blue) target tone frequency feedback trials of day 247. The patient was asked to match the target tone by modulating the normalized and weighted firing rate, and he succeeded in all but one trial of this example. Trials were completed as soon as the firing rate was held above or below the upper or lower threshold, respectively. As defined in the Materials and Methods section, these feedback blocks were performed every day of recording for training, parameter selection, and validation of the selected parameters. The grey-shaded region from -1 to 0 s depicts the time period during which the high or low target tones were presented to the patient. The horizontal line at 0.3 and 0.7 shows the lower and upper threshold, respectively. b) True positive rate vs. false positive rate of the trials in auditory neurofeedback blocks directly preceding speller blocks on days 123–462. Each circle represents one neurofeedback block; circles are jittered for better visibility. The blue insert at the bottom right corner shows the contingency table of all neurofeedback trials directly preceding speller blocks on days 123-462. In the blue insert – TP stands for true positive, i.e., up trials classified as up; FP stands for false positive, i.e., up trials classified as down; FN – stands for false negative, i.e., down trials classified as up; TN – stands for true negative, i.e., down trials classified as down; Time out denotes the trials that were unclassified.

Changes in the manuscript: Updated Figure 4 with the extended dataset

Caption - Figure 4: Overview of BCI use – a) Number of characters spelled by the patient during speller sessions whose output was rated ‘intelligible’ (rating described in text), aggregated by day. b) Characters selected per minute in ‘intelligible’ speller sessions, aggregated by day. c) Speller use during the period presented. Sessions span 135 days. Green bars represent days on which speller was used and yielded intelligible output (44 days). Yellow bars represent days on which speller use was attempted, but no intelligible output was produced (63 days). On 28 days, speller use was not attempted (red).

Changes in the manuscript: Updated Supplementary Figure S1 with the extended dataset

Caption - Supplementary Figure S1: Audio-Neurofeedback Accuracy – a) Accuracy in all neurofeedback sessions from day 106 (first-time neurofeedback speller was attempted) to day 462. The accuracy of each neurofeedback block is represented as a dot. The red dots represent neurofeedback blocks preceding a speller session. b) and c) Accuracies of neurofeedback sessions on two different days as examples. The x-axis is the time in minutes, representing the time at which a particular NF session, denoted by a black dot, was performed on that particular day, and the y-axis is the accuracy in percent.

Changes in the manuscript: Result section – line 182 to 197

“Figure 2A shows individual neurofeedback trials, including an error trial, of one representative block. Over the reported period, there were 1176 feedback sessions as shown in Supplementary Figure S1. In the 281 neurofeedback blocks preceding the speller blocks, 4936 of 5700 trials (86.6%) were correct (Figure 2B), i.e., for target tone up (higher frequency) the decision was up (a “yes” answer), and for target down (low frequency) the decision was down (a “no” answer). The difference in error rates between ‘up’ and ‘down’ trials, i.e., the fraction of trials in which the modulated tone did not match the target tone, (13.2% and 12.2%, respectively), was significant (Pearson’s χ^2 test: $p < 0.01$). The patient maintained a high level of accuracy in the neurofeedback condition throughout the reported period: on 52.6% of the days, the accuracy was at least 90% during at least one of the feedback trials blocks, i.e., the patient was able to match the frequency of the feedback to the target 18 out of 20 times. We observed considerable within-day variability of neural firing rates and hence performance of the neurofeedback classifier, necessitating manual recalibration throughout the day (see Supplementary Figure S1). In the last feedback sessions before speller sessions, the median accuracy was 90.0%, the minimum was 50.0%. In 17.1% of the sessions, accuracy was below 80.0%.”

As asked by the reviewer, in the revised manuscript, we have provided a supplementary figure that pictorially depicts the Utah array status (Supplementary Figure S2). Supplementary Figure S2b shows that a large number of channels have an acceptable impedance level (i.e., 100-800k Ω) throughout the number of days (day 1 to 462) reported in this paper.

Changes in the manuscript: New Supplementary Figure S2 depicting the state of the Utah array

Caption - Supplementary Figure S2: State of the Utah array over time – a) Depicts the channels grouped by average firing rate. The dashed, thin solid, and thick solid trace represent the number of channels between 0.5 and 2 Hz firing rate, 2 and 10 Hz firing rate, and greater than 10 Hz firing rate, respectively, from day 26 to day 462 post-implantation. b) Depicts the channels grouped by electrode impedance. The dashed, thick solid and dotted trace represents the number of channels less than 100 k Ω , greater than 100 k Ω and less than

800 k Ω , and greater than 800 k Ω impedance, respectively, from day 26 to day 462 post-implantation.

3. The statement of “one or more” neurons in the Methods should be specified precisely. There are few enough experimental days to list what was done every day, or else it would be informative to describe in detail what was most commonly done, and whether this relied on the same 1-2 channels throughout the year.

Response: We would like to thank the reviewer for the comment. As suggested by the reviewer, we have added further details on the daily experimental routine in the result section of the revised manuscript from **line 149 to 180**, as shown below.

Changes in the manuscript: Result section - Line 149 to 180

“The Results reported here include data from days 106–462 after implantation. Three of the authors frequently traveled to the patient’s home to perform communication sessions about every two weeks for 3 to 4 consecutive days until February 2020. Because of the COVID pandemic from March 2020 to June 2020, all the sessions were performed via secured remote access to the patient’s laptop. The patient’s wife performed locally all required hardware connections, and the experimenters, either UC or AT, controlled the software remotely. During the experimental period reported here, the authors UC, AT, and NB performed experimental sessions on 135 days. The patient was hospitalised due to unrelated adverse events between days 120 and 145, 163 and 172, and 212 and 223 after implantation. Each day, we started with a 10-minutes baseline recording, where the patient was instructed to rest. During this time period the experimenter ran a software program to determine the firing rate of different channels and select their parameters for the first neurofeedback session-block. Two different types of neurofeedback sessions were performed consecutively on each day, “feedback without reward” and “feedback with reward” with the goal 1) to select channels suitable for voluntary control by the patient and 2) to train the patient to control the selected channels’ spiking activity voluntarily. The first paradigm (“feedback without reward”) provided successive target tones, and the patient was asked to match the frequency of the feedback tone to the target tone. The second paradigm (“feedback with reward”) was the same. However, upon reaching and holding (during a configurable number of interactions, each interaction lasting 250 ms) the feedback tone within a predefined range around the target frequency, an additional reward sound was delivered for 250 ms, indicating successful performance to the patient. Holding the feedback tone at the high (low) end of the range for a minimum of 250 ms was then interpreted as a successful “yes” (“no”) response. After the first “feedback without reward” session, individual channels’ firing rate distributions were automatically calculated. The experimenter selected channels with differential modulation for the high and low target tones and updated the parameters for subsequent sessions. Employing this iterative procedure on each day, we performed several neurofeedback blocks within a particular day to remind the patient of the correct strategy to control the firing rate, each typically consisting of 10 high-frequency target tones and 10 low-frequency target tones presented in pseudo-random order and also to tune and validate the classifier. Typically, if the patient could match the frequency of the feedback to the target in 80% of the trials, we proceeded with the speller.”

We have also provided precise information on the number of channels used to perform speller sessions on a given day in **Supplementary Figure S3** and discussed it in the discussion section.

Changes in the manuscript: New Supplementary Figure S3 depicting the channel ID used on different days for neurofeedback and spelling.

Supplementary Figure S3 – a) Channels used for speller control over time, sorted by frequency of use. The color of the marker indicates how many channels were used simultaneously during that particular speller block. One channel was used for control in 268 blocks, two were used in 56 blocks, three in 6, and four channels were used in 2 blocks. Each day is indicated by a tick mark. b) Channel use aggregated over recording days.

4. Similarly, only successful days are described in the Results when half of the days did not result in free spelling. The graphs show that there was successful training and yes/no questions asked on many of the days where this did not work. The two datasets should be analyzed together to say whether a successful training and question experiment indicated intelligible use, or not. More examples of speech should be given from non-successful days, and there should be more analysis of the neural data and training datasets on those days to look into whether failures can be explained.

Response: We would like to thank the reviewer for this important comment and affirm that we did not just describe the successful sessions in the manuscript. In the revised manuscript, we have included the classification accuracy of all the neurofeedback sessions and

highlighted neurofeedback sessions' classification preceding a speller session in Supplementary Figure S1, as shown above. In addition, Figure 4 lists all successful and unsuccessful attempts.

Changes in the manuscript: Updated Supplementary Figure S1 with the extended dataset included all the neurofeedback sessions and not just the successful ones or ones preceding the speller session. The same is true for Figure 4.

To answer the reviewer's comment regarding whether a successful training and question experiment indicated intelligible use or not, we calculated in addition to all session results in Figure 4 the Spearman correlation between Neurofeedback task accuracy and speller intelligibility. We included the results in the "Speller sessions" section of the revised manuscript from **line 218 to 222**, as shown below. Indeed, good performance in the neurofeedback task is not a good predictor for subsequent speller success, as fewer than half of the days with attempted speller sessions, for which good NF performance was a prerequisite, led to speller output that was rated as intelligible. If there are better predictors for speller performance is part of ongoing research.

Changes in the manuscript: Speller sessions - Line 218 to 222

"There were 312 pairs of speller blocks and preceding neurofeedback blocks. The speller output was rated 0 for unintelligible, 1 for partially intelligible, 2 for intelligible. The Spearman correlation between Neurofeedback task accuracy and speller intelligibility was 0.282 ($p=4.002e-07$). The Spearman correlation between Neurofeedback accuracy and number of letters spelled was 0.151 ($p=7.671e-03$).

As suggested, we have also included more examples of "spelled" words and sentences in the result section of the revised manuscript from **line 223 to 263**, as shown below.

Changes in the manuscript: Result section- Line 223 to 263

"On the second day of free spelling, i.e., on the 107th day after implantation, the patient spelled phrases, spelled in three-time episodes, thanking NB and his team ('ERST MAL MOECHTE ICH MICH NIELS UND SEINE BIRBAUMER BEDANKEN' – 'first I would like to thank Niels and his birbaumer'). Many of the patient's communications concerned his care (e.g. 'KOP?F IMMERLQZ GERAD' – 'head always straight', day 161; 'KEIN SHIRT ABER SOCKEN' – 'no shirts but socks [for the night]', day 244; 'MAMA KOPFMASSAGE' – 'Mom head massage', day 247; 'ERSTMAL KOPFTEIL VIEL VIEL HOEH AB JETZT IMM' – 'first of all head position very high from now', day 251; 'AN ALLE MUESSEN MIR VIEL OEFTER GEL AUGENGEL' – 'everybody must use gel on my eye more often', day 254; 'ALLE SOLLEN MEINE HAENDE DIREKTEN AUF BAUBCH' – 'everybody should put my hand direct on my stomach', day 344; 'ZUM GLOTZE UND WENN BESUCHEN DA IST DAS KOPFTEIL IMMER GAAANZ RAUF' – 'when visitors are here, head position always very high' on day 461. The patient also participated in social interactions and asked for entertainment ('COME TONIGHT [to continue with the speller]', day 203, 247, 251, 294, 295, 'WILI CH TOOL BALBUM MAL LAUT HOERENZN' – 'I would like to listen to the album by Tool [a band] loud', day 245, 'UND JETWZT EIN BIER' – 'and now a beer', day 247 (fluids have to be inserted through

the gastro-tube), 251, 253, 461. He even gave suggestions to improve his speller performance by spelling 'TURN ON WORD RECOGNITION' on day 183, 'IS IT EASY BACK ONCE CONFIRMATION' on Day 253, 'TELL ALESSANDRO I NEED TO SAVE EDIT AND DELETE WHOLE PHRASES AND ALL OF THAT INTO THE LIST WHERE (patient's son name) on day 295, 'WHY CANT YOU LEAVE THE SYSTEM ON. IFIND THAT GOOD' on day 461, in English as the patient knew that the experimenter UC and AT are not native German speakers and mostly spoke in English with the patient. On day 247 he gave his feedback on speller as, 'JUNGS ES FUNKTIONIERT GERADE SO MUEHELOS', - 'boys, it works so effortlessly'. The patient expressed his desire to have different kind of food in his tube as, 'MIXER FUER SUPPEN MIT FLEISCH' – 'instructed his wife to buy a mixer for soup with meat' on day 247'; 'GULASCHSUPPE UND DANN ERBSENSUPPE' – 'Gulash soup and sweet pea soup' on day 253; 'WEGEN ESSEN DA WIRD ICH ERST MAL DES CURRY MIT KARTOFFELN HABEN UND DANN BOLOGNA UND DANN GEFUELLTE UND DANN KARTOFFELN SUPPE' – 'for food I want to have curry with potato then Bolognese and potato soup on day 462. He interacted with his 4 years old son and wife, '(son's name) ICH LIEBE MEINEN COOLEN (son's name) – 'I love my cool son' on day 251; '(son's name) WILLST DU MIT MIR BALD DISNEYS ROBIN HOOD ANSCHAUEN'- 'Do you want to watch Disney's Robin Hood with me' on day 253; 'ALLES VON DEN DINO RYDERS UND BRAX AUTOBAHNUND ALLE AUFZIEHAUTOS' – 'everything from dino riders and brax und cars' on day 309; '(son's name) MOECHTEST DU MIT MIR DISNEYS DIE HEXE UND DER ZAUBERER ANSCHAUEN AUF AMAZON' – 'would you like to watch Disney's witch and wizard with me on amazon' on day 461; 'MEIN GROESSTER WUNSCH IST EINE NEUE BETT UND DAS ICH MORGEN MITKOMMEN DARF ZUM GRILLEN' – 'My biggest wish is a new bed and that tomorrow I come with you for barbecue' on day 462.”

We have also updated the Figure 1 to explicate the experimental process using a flow chart as shown below in Figure 1b.

Changes in the manuscript: Updated Figure 1

Caption – Figure 1: a) Experimental setup. Two microelectrode arrays were placed in the precentral gyrus and superior frontal gyrus (insert, L: left central sulcus, A-P: midline from anterior to posterior). An amplifying and digitizing headstone recorded signals through a percutaneous pedestal connector. Neural signals were pre-processed on a Neural Signal Processor and further processed and decoded on a laptop computer. b) Daily sessions began with Neurofeedback training. If the performance criterion was reached, the patient proceeded to speller use. If the criterion was not reached, parameters were re-estimated on neurofeedback data, and further training was performed. c) Schematic representation of auditory neurofeedback and speller. Action potentials were detected and used to estimate

neural firing rates. One or several channels were selected, their firing rates normalized and mixed (see Online Methods). Options such as letter groups and letters were presented by a synthesized voice, followed by a response period during which the patient was asked to modulate the normalized and mixed firing rate up for a positive response and down for a negative response. The normalized rate was linearly mapped to the frequency of short tones that were played during the response period to give feedback to the patient. The patient had to hold the firing rate above (below) a certain threshold for typically 500ms to evoke a “Yes” (“No”) response. Control over the neural firing rates was trained in neurofeedback blocks, in which the patient was instructed to match the frequency of target tones.

Reviewer #2 (Remarks to the Author):

1. In this study, the authors demonstrate a system that enables volitional communication for an individual in a completely locked-in state, which has not been demonstrated before. Unlike systems developed for those in a locked-in state which rely on visual feedback, this system utilizes auditory feedback to assist the user in spelling words, thus eliminating the need for a functional ocular system to communicate.

The authors' conceptual approach to designing a communication system for a user that is completely locked-in makes sense as long as the ocular system cannot be used for feedback at all. It is unclear from the article, but if visual feedback could be presented without requiring eye movements, perhaps an approach like Vansteensel et al., 2016 would be preferable since the communication rates are higher. As it stands, the article also lacks sufficient analysis and quantification of data that led to design decisions of the system and performance outcomes of the system. With the inclusion of additional analyses and justification of auditory feedback usage, this paper presents an advance in developing communication tools for people in a completely locked-in state. Please see my specific comments below.

Response: We would like to thank the reviewer for the comment and agree with the reviewer that with intact vision, a BCI like the one presented by Vansteensel et al. 2016, cited and acknowledged in our paper, is preferable and faster. As mentioned earlier, this individual lacked reliable voluntary eye-movement control and, consequently, was unable to use an eye-tracker for communication. The patient was also ultimately unable to use a non-invasive eye-movement-based computerized communication system. This patient in CLIS also lost the ability to open and move his eyes voluntarily and therefore his eyes are closed. Thus, a BCI employing visual feedback would be very difficult if not impossible to implement with such an individual. Hence, we aimed to establish communication with the patient independent of his vision. As mentioned in response to comment 2 of reviewer 1 we have updated the revised manuscript's introduction section accordingly.

2. Line 123: Did the authors attempt to establish communication in between days 1 and 86? I would imagine that the day after implantation would not be an accurate representation of the participant's brain state as the swelling would need to subside. Were the tones and reward signals played to participants before neurofeedback

training? Did they elicit any neural response? It would be great to see plots of the tone related spiking rates before and after starting the neurofeedback task on day 86.

Response: We would like to thank the reviewer for the comment. From day 1 to day 86, control of firing rate was tried by instructing the patient to use differential imagery strategies and eye-movements as done before entering the completely locked-in state, none of them with any success. Before the initiation of the neurofeedback procedure on day 86, none of the feedback tones and reward tones were presented to the patient. Because the feedback procedure immediately after the first attempt resulted in successful firing control and spelling, we did not test the effects of tone alone presentation on firing rate to avoid extinction or interference of the employed successful strategy of the patient. However, the animal literature after Eberhard Fetz's (1969) pioneering study of spike control with neurofeedback suggests a clear effect of reward and feedback on spike firing independent of the physical characteristics of the feedback signal and the reward. The reward used in the animal literature consisted of food, while in the many reports of the non-invasive human neurofeedback literature on EEG neurofeedback, usually visual or verbal reward (i.e., smiling face) was given. Again, the non-invasive neurofeedback research (see Birbaumer et al. 2013 for review) demonstrated control of the neural signals independent feedback and reward's physical nature. We have updated the result section of the revised manuscript from **line 133 to 138**, as shown below.

Changes in the manuscript: Result section- Line 133 to 138

“One day after the implantation, attempts were initiated to establish communication. The patient was asked to use his previous communication strategy employing eye movements to respond to questions with known “yes” and “no” answers, which did not result in a classifiable neural signal. Passive movements of the patient's right fingers, thumb, and wrist evoked consistent neural responses on several electrodes. However, when we then instructed the patient to attempt or imagine hand, tongue, or foot movements, we could not detect consistent responses.”

3. Furthermore overall brain mapping showing the SRMs for all channels should be shown. The authors rely on one or two channels for their system despite having 128 channels to work with. Many decoders with inputs from the motor cortex rely on many channels to achieve high performance, why was a multi-channel approach not taken here? In general, it seems that the reader is left in the dark as to the characteristics of the neural responses (or lack thereof) that are observed corresponding to various behavior attempts and stimuli.

Response: We would like to thank the reviewer for the comment. After the establishment of successful communication after day 86, the patient utilized similar channels from the Supplementary Motor Cortex array (mainly electrode 21 and neighboring electrodes by demonstrating differential control of the feedback tones during the neurofeedback sessions before spelling. All other 120 channels did not result in differential control between “select” (“yes”) and “no-select” (“no”) trials during the neurofeedback procedure for most sessions and days of recording. Because the neurofeedback procedure was the prerequisite for successful communication after 86 days of failure, a multichannel decoding algorithm was

not attempted at that time. This decision was reasonable concerns based on learning principles (Birbaumer et al., 2013) that modifications in the employed decoding strategy might decrease spelling performance and impair the only means of communication of the patient. Also, as shown above in Supplementary Figure S3, in response to comment number 2 of reviewer 1 only a fraction of channels showed significant firing rates. We have updated the discussion section of the revised manuscript from **line 271 to 284**, as shown below accordingly.

Changes in the manuscript: Discussion section - Line 271 to 284

“In all blocks, measurable spike rate differentiation between “yes” and “no” during the neurofeedback trials and “select” and “no select during speller blocks appeared in only a few channels in the SMA (supplementary motor area) out of all active channels, as shown in Supplementary Figure S3a. After the establishment of successful communication after day 86, similar channels from the Supplementary Motor Cortex array were used for communication sessions with the patient, as shown in Figure S3b. Mainly electrode 21 and neighbouring electrodes were used, which demonstrated differential control of the feedback tones during the neurofeedback sessions before spelling. Because the neurofeedback procedure was the prerequisite for successful communication after 86 days of failure, a multichannel decoding algorithm was not implemented following our clinical judgment based on learning principles (Birbaumer et al., 2013) that such a substantial change in the procedure might impede or extinguish the successful control of the patient and spelling. Thus, for speller, only one to four channels were used for control, as shown in Supplementary Figure S3.”

4. Line 123: Could the authors provide more detail on the lack of neural activations for attempted hand movements and contrast their findings with Freudenburg et al., 2019 (Frontiers in Neuroscience)?

Response: We would like to thank the reviewer for the comment. As stated above, in response to comment number 2 of reviewer 2 attempted motor attempts did not result in a reliably classifiable signal which could be used for communication. The reason for this result is a matter of investigation and out of the scope of this paper where our main goal is to show that a patient without any means of communication and without acute vision used an auditory neurofeedback strategy for communication. Additional analyses of motor attempt and somatosensory stimulation trials are part of ongoing research.

5. Line 173: Could the authors provide the information transfer rate for their results? This will help establish a performance baseline for futures studies. Also, including the character error rate and word error rate when in spelling mode would be helpful.

Response: We would like to thank the reviewer for the comment. Based on the reviewer’s comment, we updated Figure 4 with the number of characters spelled and the number of characters spelled per minute by the patient during every speller session. We have updated the discussion section of the revised manuscript from **line 285 to 288**, as shown below. A good and objective estimate of the speller information transfer rate, character error rate, and word error rate is difficult to obtain in this case, as only few copy-spelling blocks of short duration were obtained. To derive these metrics from free spelling blocks, the experimenters’

judgment would have to be relied upon. Finally, we hesitate to state ITR for neurofeedback trials due to the inherent differences in timing with the speller.

Changes in the manuscript: Discussion section - Line 285 to 288

“As shown in Figure 4, the patient spelled a different number of characters on different days. He spelled only under 100 characters on some days, while on other days, he spelled more than 400 characters. Despite the huge variation in the number of characters spelled, the number of characters spelled per minute was close to 1.00 character per minute.”

6. Line 346: The authors use two forms of neurofeedback (tones and a reward sound), but there is no analysis or discussion on how these forms of feedback influenced the task. Are both forms of feedback necessary, or could the task be performed with the reward feedback alone? How important is linearizing the tone response as opposed to taking a more binary classification approach?

Response: We would like to thank the reviewer for the comment. In order to facilitate the learning of the system, i.e., what mental activity translates to modulation of firing rates, we provided gradual feedback. In addition, we provided feedback on task accomplishment because the holding of the activity for a certain amount of time was important. However, as mentioned in response to comment 2 of reviewer 2 we did not perform any systematic investigation of how these feedback forms influenced the task. Therefore, it is difficult to speculate on whether these two forms of feedback were necessary or whether the task could have performed with the reward tone alone. The reason for not doing so was the above-mentioned concerns of patient's success extinction. However, a large literature on learning principles over the last 100 years clearly shows in motivated human participants that either feedback or reward is sufficient for learning behavioral (and physiological) control, and the combination of both usually has no substantial additional effects (see Birbaumer et al., 2013 for review).

7. Supp. Figure 1: Plotting the mean/median performance instead of the maximum performance is preferable. Given the high performance variability, the maximums do not seem meaningful. On that note, why is the variance across trial sets so high? It may be beneficial to the read to see mean/std event related spiking rate plots of selected channels during the tone and spelling tasks. Would using a multi-channel system help stabilize the performance of the system?

Response: We would like to thank the reviewer for the comment. Supplementary Figure S1 shows each neurofeedback block's accuracy, even as the model parameters are still adapted and the model learned by the participant. Instead of highlighting the daily maximum, we now show the neurofeedback accuracy just before speller blocks (same blocks as in Fig 3B, with the addition of blocks from the period until 20 July 2019 for which the calculation of TPR/FPR was impossible due to the way events were recorded, but for which correct and incorrect trials could still be determined). Our system was capable of integrating multiple channels, and multiple channels were used in several sessions (see Supplementary Figure S3). However, we found that on many days, only one channel's firing rate was significantly and consistently modulated during the task. We have since extended the BCI paradigm to use other features as well, for which we see more consistent modulation and stability. These data will be treated in a forthcoming publication.

8. Figure 1B and 4 captions do not adequately explain the figures. The caption should describe all aspects of figures.

Response: We apologize for that. The figure captions have now been adequately explained, as shown below. Old Figure 4 is now Figure 3.

Changes in the manuscript: Caption of Figure 1 and Figure 3

Caption – Figure 1: a) Experimental setup. Two microelectrode arrays were placed in the precentral gyrus and superior frontal gyrus (insert, L: left central sulcus, A-P: midline from anterior to posterior). An amplifying and digitizing headstage recorded signals through a percutaneous pedestal connector. Neural signals were pre-processed on a Neural Signal Processor and further processed and decoded on a laptop computer. b) Daily sessions began with Neurofeedback training. If the performance criterion was reached, the patient proceeded to speller use. If the criterion was not reached, parameters were re-estimated on neurofeedback data, and further training was performed. c) Schematic representation of auditory neurofeedback and speller. Action potentials were detected and used to estimate neural firing rates. One or several channels were selected, their firing rates normalized and mixed (see Online Methods). Options such as letter groups and letters were presented by a synthesized voice, followed by a response period during which the patient was asked to modulate the normalized and mixed firing rate up for a positive response and down for a negative response. The normalized rate was linearly mapped to the frequency of short tones that were played during the response period to give feedback to the patient. The patient had to hold the firing rate above (below) a certain threshold for typically 500ms to evoke a “Yes” (“No”) response. Control over the neural firing rates was trained in neurofeedback blocks, in which the patient was instructed to match the frequency of target tones.

Caption – Caption – Figure 3: Example of letter selection during a free spelling block – a) Firing rate of the channels 25 and 100 used for “yes”/“no” classification on day 108. b) Normalized firing rate and the speller state during the same 90 second period of a speller block. “Yes” / “no” / timeout decisions are marked by vertical lines and the option selected in green and not selected in red. This example is part of the phrase “DEKUBITUS PO ER SOLL ARME MAXIMAL”, referring to bedsore and instructing the aide to change arm position.

9. Figure 2: The figure focuses primarily on duration of the BCI usage, but perhaps a better aspect of the data is to highlight the performance of the system on each day, especially the speller. Instead of using a subjective three-way classification for speller performance, an objective measure like character error rate should be used. This figure could be two panels where an adaptation of supp. Figure 1 is B.

Response: We would like to thank the reviewer for the comment. We have modified the Figure in the revised manuscript as stated above in response to comment 2 of reviewer 1. We have added three main information to elucidate the patient's speller performance a) the number of characters spelled by the patient during a particular speller session, b) the characters per minute selected by the patient during a particular speller session, and c)

speller's output rating. This information in our opinion best highlights the performance of the system each day, as character error rate in a free spelling paradigm also relies on the experimenter's judgment of intended communication. We have also added text on patient's speller performance in the revised manuscript from **line 210 to 218**, as shown below.

Changes in the manuscript: Speller sessions section - Line 210 to 218

“Over the reported period, out of 135 days, speller sessions were attempted on 107 days, while on the remaining 28 days use of the speller was not attempted because the neurofeedback performance criterion was not reached. The patient produced intelligible output, as rated independently by three observers, on 44 of 107 days when the speller was used (Figure 4). On average, 121 minutes were spent spelling and the average length of these communications was 131 characters per day. The patient's intelligible messages comprised 5747 characters produced over 5338 minutes, corresponding to an average rate of 1.08 characters per minute. This rate varied across blocks (min/median/max: 0.2/1.1/5.1 characters per minute). Over the reported period, there was no apparent trend in spelling speed.”

10. Figure 3A. There is no indication of what the grey bar represents in the caption, I'm assuming it is the tone presentation period

Response: The gray bar is now explained. Please see the response to comment number 2 of reviewer 1.

References:

Fetz, E.E., 1969. Operant conditioning of cortical unit activity. *Science*, 163(3870), pp.955-958.

Birbaumer, N., Ruiz, S. and Sitaram, R., 2013. Learned regulation of brain metabolism. *Trends in cognitive sciences*, 17(6), pp.295-302.

Reviewers' comments:

Reviewer #1 (Remarks to the Author):

The manuscript is much improved. All of my concerns have been resolved by the authors' edits.

Reviewer #2 (Remarks to the Author):

My general concerns is that the claims are too strong (starting with the title), standard metrics like ITR not reported (despite reasonable request), and more information needed on the array recordings. The performance is not very good, and may actually be interpreted opposite of what is claimed (i.e. bci decoding in CLIS patients is poor given what has been previously published by Braingate).

SPECIFIC COMMENTS:

Title is misleading, and it should definitely be changed. Can suggest something like "Spelling interface using intracortical signals in a completely locked-in patient enabled via auditory neurofeedback training".

Please provide ITR, not sure why there is so much push back there.

Can the authors add to Supplementary Figure S3 another panel showing the location of the two arrays on the brain (can even re-use the illustration from Figure 1), and then in panel B of that figure where you list the most important channel IDs, add another row to these x-axis labels that indicates which of the two arrays each channel came from? This will help the readers easily see the distribution of important channels in the two arrays.

Line 186: Please also report the accuracy from all neurofeedback blocks regardless of whether they preceded a spelling block.

The article in its current form does not provide the reader with any indication of how accurate the speller is. Therefore, it is vital for the authors to include a character/word error rate per session. Additionally, many BCI studies include an information transfer rate (ITR) metric. Including ITR for the spelling portion of this study is important so that the study can be put into context with current and future work. If necessary, free spelling blocks can be used in addition to copy spelling to calculate ITR and character error rates as long as the authors' rationale is thoroughly explained in the methods. Without ITR and character error rates, the article is insufficient for publication as these are important metrics for understanding the results.

Could the authors provide a visualization of the location of the arrays? Perhaps with an MRI? Furthermore, labeling the most used channels would also be beneficial.

In lines 291–313, the authors defend the assessment of CLIS for their participant, even though they state that electrooculography (EOG) revealed some significant differences between the "yes" and "no" binary classification conditions. The authors acknowledge that this finding conflicts with the diagnosis, suggest that it is the fault of the EOG-based BCI system, and then cite the lamentation of the family (regarding the participant's inability to communicate) to further defend the CLIS diagnosis. However, my takeaway is that this is purely a technological or implementation issue, and that if the EOG-based BCI system was better (or more personalized to the study participant), then it would have been functional. If post-hoc analyses reveal significant differences between "yes" and "no" for the maximum, mean, and variance features, then this should indicate that EOG-based communication is possible, at least to some degree.

All this being said, the authors cannot be expected to develop and assess a refined EOG-based BCI for the participant. Instead, I think it would be helpful to have the authors comment a bit more about the CLIS diagnosis in general. I find it unsettling that this CLIS diagnosis was clearly due to

a limitation in the performance of the available EOG-based BCIs. Do they think that other individuals with CLIS might be capable of exhibiting some level of EOG-based differentiability control and just have the available decoding interfaces failing them? Are there studies involving CLIS individuals that characterize EOG signals in those cases? I suggest that the authors add more to the discussion (or in a Supplementary Text) about what they think EOG activity might look like in other CLIS participants, citing additional literature as necessary. It also seems reasonable for the authors to explicitly acknowledge in their Discussion section that the CLIS diagnosis is very much tied to existing assistive technology and cannot be assessed independent of existing decoding systems. The authors' addition of the current CLIS EOG findings to the existing literature (if there is any) could inform future non-invasive communication interfaces (and CLIS diagnosis procedures).

- Figure 1: In the letter groups during the spelling task, why were some of the letters out of order?

For example, "E A D C B F" instead of "A B C D E F"?

- Figure 1: In the caption, please explicitly state that the blue and orange items depict two channels, even though more (or fewer) channels might be used at certain times (similar to how those lines are labeled in Figure 3).

- Did the authors do any tests to assess what baseline "normalized neural activity" would be? That is, what would this normalized level be if the user was just at rest and not actively trying to modulate it? Would a middle-level tone be output in the neurofeedback task?

- Line 299: Please define the acronym "EOG" when introducing it for the first time.

Reviewers' comments:

Reviewer #1 (Remarks to the Author):

The manuscript is much improved. All of my concerns have been resolved by the authors' edits.

Response: Thank you.

Reviewer #2 (Remarks to the Author):

My general concerns is that the claims are too strong (starting with the title), standard metrics like ITR not reported (despite reasonable request), and more information needed on the array recordings. The performance is not very good, and may actually be interpreted opposite of what is claimed (i.e. bci decoding in CLIS patients is poor given what has been previously published by braingate).

Response: Indeed, the system does not work very reliably. However, this has to be seen in context. All BrainGate participants had the ability to move and communicate in one form or another. We do not know whether the difficulties of maintaining stable control is a result of ALS and the associated complete loss of movement: it is likely that the inability to move at all also affects the ability to generate stable and repeatable motor cortex activity (despite the diagnosis of no cortical atrophy before implantation). Moreover, the patient's cognitive, psychological, and attentional state may contribute to the difficulties of achieving reliable control. As there is no other report in the literature demonstrating communication in a patient in such an advanced state, it is hard to know and control all the aspects that need to be addressed. These issues should be addressed in further research.

Change in the manuscript:

Added sentence in discussion to compare with late stage ALS P300 speed (which is much slower), visual SSVEP (which is comparable) and intracortical arrays in much more able patients (which is much faster)

SPECIFIC COMMENTS:

Title is misleading, and it should definitely be changed. Can suggest something like "Spelling interface using intracortical signals in a completely locked-in patient enabled via auditory neurofeedback training".

Response: Title changed as suggested.

Line 186: Please also report the accuracy from all neurofeedback blocks regardless of whether they preceded a spelling block.

Response: Added sentence to legend of S1, showing all the NF accuracies in the revised manuscript.

Please provide ITR, not sure why there is so much push back there. The article in its current form does not provide the reader with any indication of how accurate the speller is. Therefore, it is vital for the authors to include a character/word error rate per session. Additionally, many BCI studies include an information transfer rate (ITR) metric. Including ITR for the spelling portion of this study is important so that the study can be put into context with current and future work. If necessary, free spelling blocks can be used in addition to copy spelling to calculate ITR and character error rates as long as the authors' rationale is thoroughly explained in the methods. Without ITR and character error rates, the article is insufficient for publication as these are important metrics for understanding the results.

Response: We addressed this point in a revised version of the manuscript, even though we think that the ITR estimate will be inaccurate (different timing between neurofeedback trials, where ground truth is known, and speller, where ground truth is unknown and can only be assumed, and only a few copy speller sessions with known target words). It is important to state again that the implantation and use of the speller was done on a compassionate use basis, as the participant does not have another means of communication. All sessions with the patient were therefore aiming at restoring communication (by providing a free speller) rather than collecting ground truth research data.

Change in the manuscript:

Added calculation of ITR in methods and results (around 5 bits/min on average).

Can the authors add to Supplementary Figure S3 another panel showing the location of the two arrays on the brain (can even re-use the illustration from Figure 1), and then in panel B of that figure where you list the most important channel IDs, add another row to these x-axis labels that indicates which of the two arrays each channel came from? This will help the readers easily see the distribution of important channels in the two arrays.

Could the authors provide a visualization of the location of the arrays? Perhaps with an MRI? Furthermore, labeling the most used channels would also be beneficial.

Response: Figure 1 contains the actual array locations as reconstructed from the MRI and intra-surgical photographs. We included the requested channel locations in supplementary figure 3.

Change in the manuscript:

Added panel in supplementary figure 3

In lines 291–313, the authors defend the assessment of CLIS for their participant, even though they state that electrooculography (EOG) revealed some significant differences between the “yes” and “no” binary classification conditions. The authors acknowledge that this finding conflicts with the diagnosis, suggest that it is the fault of the EOG-based BCI system, and then cite the lamentation of the family (regarding the participant’s inability to communicate) to further defend the CLIS diagnosis. However, my takeaway is that this is purely a technological or implementation issue, and that if the EOG-based BCI system was better (or more personalized to the study participant), then it would have been functional. If post-hoc analyses reveal significant differences between “yes” and “no” for the maximum, mean, and variance features, then this should indicate that EOG-based communication is possible, at least to some degree.

All this being said, the authors cannot be expected to develop and assess a refined EOG-based BCI for the participant. Instead, I think it would be helpful to have the authors comment a bit more about the CLIS diagnosis in general. I find it unsettling that this CLIS diagnosis was clearly due to a limitation in the performance of the available EOG-based BCIs. Do they think that other individuals with CLIS might be capable of exhibiting some level of EOG-based differentiability control and just have the available decoding interfaces failing them? Are there studies involving CLIS individuals that characterize EOG signals in those cases? I suggest that the authors add more to the discussion (or in a Supplementary Text) about what they think EOG activity might look like in other CLIS participants, citing additional literature as necessary. It also seems reasonable for the authors to explicitly acknowledge in their Discussion section that the CLIS diagnosis is very much tied to existing assistive technology and cannot be assessed independent of existing decoding systems. The authors’ addition of the current CLIS EOG

findings to the existing literature (if there is any) could inform future non-invasive communication interfaces (and CLIS diagnosis procedures).

Response: Assessment of CLIS. Locked-in Syndrome (LIS) criteria defined by the American Congress of Rehabilitation Medicine (ACRM) includes quadriplegia and eye movements or blinking as the main mode of communication (1). Total or Complete LIS has been defined as ‘Total immobility and inability to communicate’, including loss of eye movements (2). This is the definition we operationalized, as this is of practical relevance for the person in a locked-in state and their caregivers. We additionally considered commercial assistive technology as means by which communication could be achieved. The participant at implantation had lost voluntary movements and the ability to control any such devices (eye trackers, switches; last use in August 2017). To our knowledge, there is no commercial device to restore communication using EOG or EEG signals for CLIS patients. We, therefore, deem the CLIS diagnosis based on published criteria and available technology valid and relevant. The participant also took part in a previous study to assess EEG and EOG (ref 22 in the article). Communication using these EOG signals was possible for several months, but – as shown as supplementary data in the manuscript – differences in the EOG between ‘yes’ and ‘no’ conditions decreased steadily. While there were occasional significant differences between conditions, it is important to highlight that these differences are only detectable over tens of trials and are thus unsuitable for a practical means of communication. The deterioration of the EOG signals over time also indicated that differences were likely to be extinguished as a matter of time. With data from the EOG experiments, we, therefore, attempted to give supporting quantitative evidence for the condition.

We agree that the CLIS diagnosis is tied to existing technology, therefore we have added a sentence to highlight this definition problem.

The literature about CLIS patients is very poor and there is no study - to the best of our knowledge - that extensively investigate the residual muscle activity of these patients. The only studies that provide metrics for the muscle activity have been performed in previous studies of the authors of the article, in particular the EOG-BCI study (ref 22 in the article) provide measures of EOG amplitude in patients in transition from LIS to CLIS, nonetheless the ocular activities of the four included patients (one is the same patient of this article) vary between $\pm 200\mu V$ and $\pm 40\mu V$, while in the article the EOG amplitude of the patient when he communicate through EOG for the last time was in the range $\pm 20\mu V$. We added two sentences to highlight these points, but we believe that an extensive study of CLIS EOG activity is needed for future non-invasive communication interfaces.

• Figure 1: In the letter groups during the spelling task, why were some of the letters out of order? For example, “E A D C B F” instead of “A B C D E F”?

Response: Added sentence in Methods. Review/confirm accuracy.

- Figure 1: In the caption, please explicitly state that the blue and orange items depict two channels, even though more (or fewer) channels might be used at certain times (similar to how those lines are labeled in Figure 3).

Response: Added clarification in legend for Fig 1.

- Did the authors do any tests to assess what baseline “normalized neural activity” would be? That is, what would this normalized level be if the user was just at rest and not actively trying to modulate it? Would a middle-level tone be output in the neurofeedback task?

Response: Neural activity during baseline is not constrained by a task, and no feedback is given. We therefore do not have an expectation of the normalized signal to assume any particular state. Figure 3 shows a representative period during a speller session. Baseline periods (white background) show different patterns: residual state from previous trial (trial starting at 995s), anticipation of upcoming correct response (trials starting at 1000s or 1027s), or state opposite of upcoming response (982s). We also observe normalized activity during baseline in the 'undecided region', e.g. in trial starting at 1032s.

- Line 299: Please define the acronym “EOG” when introducing it for the first time.

Response: Done.

REVIEWERS' COMMENTS

Reviewer #2 (Remarks to the Author):

The authors have addressed nearly all of my concerns adequately. One remaining issue is that the results here are highly discrepant with what has been described in BrainGate. For example, last summer a subject achieved handwriting speed for world record bit rate (possibly faster than normal handwriting). This is the issue that keeps coming up about the limited performance shown in the current paper.

Readers will definitely expect some speculation from the researchers about why there is such huge difference in performance. One of the most important takeaways for me is precisely the point the authors made in the introduction, which is that CLIS may be very different than other forms of paralysis.

The authors need to be more clear about addressing this in the discussion about the differences in performance from the Braingate cases, as it suggests the Braingate participants are more fully functional than most appreciate.

REVIEWERS' COMMENTS

Reviewer #2 (Remarks to the Author):

The authors have addressed nearly all of my concerns adequately. One remaining issue is that the results here are highly discrepant with what has been described in BrainGate. For example, last summer a subject achieved handwriting speed for world record bit rate (possibly faster than normal handwriting). This is the issue that keeps coming up about the limited performance shown in the current paper. Readers will definitely expect some speculation from the researchers about why there is such huge difference in performance. One of the most important takeaways for me is precisely the point the authors made in the introduction, which is that CLIS may be very different than other forms of paralysis. The authors need to be more clear about addressing this in the discussion about the differences in performance from the Braingate cases, as it suggests the Braingate participants are more fully functional than most appreciate.

Response: We are pleased to read that the reviewer finds our response appropriate. We agree with the new comment of the reviewer, and to address that, we revised the discussion section substantially, as shown below.

Changes in the manuscript:

Line 289 – 296:

“In addition, after this failure, we attempted a neural feedback approach, based on learning principles, capitalizing on the observation that neural firing rates could be used to achieve levels suitable to make yes/no choices. For the speller sessions, only one to four channels were used for control, as shown in Supplementary Figure S3. There was insufficient time to explore other decoding approaches, and we wanted to establish that communication was feasible at all in CLIS. We cannot explain why the other electrodes did not provide modulation suitable for multichannel decoding. Perhaps with further sessions and other strategies, not possible in this experiment, we might have identified faster, or more accurate approaches.”

Line 349 – 376:

“In this study, communication rates were much lower compared to other studies using intracortical arrays, which include communication with a point-and-click screen keyboard⁷⁻⁹, and decoding of imagined handwriting in a spinal cord injured patient²⁷. People with ALS and not apparently completely locked-in have been able to use multi neuron-based decoders for more rapid communication than seen here. The differences might be both technical - a failure of the electrode - or biological, i.e., related to the disease state. While the multielectrode array (MEA) used here typically shows some variable level of degradation in the quality and number of recordings over time, such arrays reportedly provide useful signals for years^{28,29}.

Our MEA retained impedances in the useful range across the entire experimental period of this study (Fig S2) and neurons were recorded on many channels suggesting that the loss of recordable neural waveforms cannot explain performance differences. The most striking difference in the present data was the inability

to record neurons that modulated with the participant's volitional intent. This could be the result of the disease processes on the neurons themselves or the protracted loss of sensory-motor input itself. The observation that control was intermittent may reflect changes in neural connectivity or the ability to be activated. Lack of any somatic sensory feedback, especially that from muscles, might impede voluntary modulation of neural activity. Participants with ALS enrolled in previous trials apparently had at least some residual voluntary control of muscles, whereas this participant had lost all control by the time of implantation. Additionally, advanced ALS may have led to cognitive or affective changes such as shortened attention span or modified motivational systems that may have made it difficult to achieve reliable modulation of large numbers of neurons (dozens to hundreds) achieved in other ALS participants with similar BCI systems implanted. Altered cortical evoked response amplitudes and latencies³⁰ seen in this individual may be a reflection of these abnormal states. CLIS patients with ALS show highly variable and often pathological neurophysiological signatures³¹ such as heterogeneous sleep-waking cycles³² that may also affect the ability to engage neurons. Lastly, auditory cues may engage motor processes that will activate neurons in frontal areas outside of the motor cortex³³, which may have contributed to the changes with the auditory task used here."